# Squeezed light from an oscillator measured at the rate of oscillation

Christian Bærentsen [1], Sergey A. Fedorov [1] ✉, Christoffer Østfeldt[1], Mikhail V. Balabas [1], Emil Zeuthen [1] & Eugene S. Polzik [1] ✉

Sufficiently fast continuous measurements of the position of an oscillator approach measurements projective on position eigenstates. We evidence the transition into the projective regime for a spin oscillator within an ensemble of $2 \times 10^{10}$ room-temperature atoms by observing correlations between the quadratures of the meter light field. These correlations squeeze the fluctuations of one light quadrature below the vacuum level. When the measurement is slower than the oscillation, we generate $11.5^{+2.5}_{-1.5}$ dB and detect $8.5^{+0.1}_{-0.1}$ dB of squeezing in a tunable band that is a fraction of the resonance frequency. When the measurement is as fast as the oscillation, we detect 4.7 dB of squeezing that spans more than one decade of frequencies below the resonance. Our results demonstrate a new regime of continuous quantum measurements on material oscillators, and set a new benchmark for the performance of a linear quantum sensor.

Projective, or von Neumann, measurements collapse the observed quantum system on eigenstates of a Hermitian operator, while more general measurements, described by positive operator-valued measures, collapse the system on states from an overcomplete set[1]. A gradual transition between the two situations can be realized in continuous measurements using meter fields, a canonical example of which is an optical interferometric measurement of the lab-frame position of a harmonic oscillator[2]. Such measurements are associated with mechanical resonators[3], collective atomic spins[4,5], ferromagnetic solid-state media[6], single molecules[7], or density waves in liquids[8], that are linearly probed by traveling optical or microwave fields. The boundary between generalized and von Neumann measurements occurs at a certain value of the measurement rate[9–11]. When the rate is slower than the oscillation, measurements with the meter in the vacuum state project the oscillator on coherent states. When the rate is faster than the oscillation, they project the oscillator on position-squeezed states.

The output state of the meter field changes with the measurement regime as much as the oscillator state itself[10]. The quadratures of the meter are correlated to the extent that their fluctuations can be below the vacuum level at certain quadrature angles[12,13]. In the slow-measurement regime, due to the time-averaged response of the oscillator to the measurement backaction, the correlations and the

associated squeezing exist in a narrow frequency band near the resonance. When the measurement is faster than the oscillation, the oscillator responds to the backaction nearly instantaneously, and the correlations and squeezing are broadband. The detection of such broadband squeezing of light implies observing the backaction-driven motion of the oscillator at frequencies well below the resonance frequency, which is a necessary condition for position squeezing[10].

The squeezing of the meter light is both a valuable quantum resource and a figure of merit for the purity of the light–oscillator interaction. In the slow regime, we realize a measurement of a collective spin of a room-temperature atomic ensemble at a rate fifteen times higher than the rate of thermal decoherence. The generated squeezing of the meter light reaches $11.5^{+2.5}_{-1.5}$ dB at the output of the cell, substantially exceeding the squeezing demonstrated previously using collective atomic spins[14–16], optomechanical cavities[17–19], levitated nanoparticles[20,21], and compact on-chip sources utilizing material nonlinearity[22], while approaching the results achievable using bulk nonlinear crystals[23]. In the fast-measurement regime, we detect broadband squeezing in a bandwidth of several MHz while keeping the backaction-imprecision product[24] within 20 % from the value saturating the Heisenberg uncertainty relation. These results enable new regimes for sensing surpassing the standard quantum limit[25,26], tests of uncertainty relations for past quantum states[27,28], quantum control of

---

[1]Niels Bohr Institute, University of Copenhagen, Copenhagen, Denmark. ✉e-mail: sergey.fedorov@nbi.ku.dk; polzik@nbi.dk

material oscillators[29–33], and links between collective spins and other material systems[16,34–36].

## Results

### Measurements of spin oscillators

Linearly polarized light traveling through an oriented atomic medium (as illustrated in Fig. 1a, b) continuously measures the projection of the total spin on the propagation direction, $\hat{J}_z$, via polarization rotation. This measurement acts back on the spin via quantum fluctuations of optical torque. When the input light is in a strong coherent state, and the spin satisfies the Holstein–Primakoff approximation[37], the process can be described in terms of linearly coupled pairs of canonically conjugate position and momentum variables. The canonical variables of the spin, $\hat{X}_S$ and $\hat{P}_S$, are the normalized projections defined as $\hat{X}_S = \hat{J}_z/\sqrt{\hbar\langle J_x\rangle}$ and $\hat{P}_S = -\hat{J}_y/\sqrt{\hbar\langle J_x\rangle}$, which satisfy the commutation relation $[\hat{X}_S, \hat{P}_S] = i$. The variables of the light, $\hat{X}_L$ and $\hat{P}_L$, are the quadratures proportional to the amplitude and phase differences between the circularly polarized components, respectively. Their commutator is $[\hat{X}_L(t), \hat{P}_L(t')] = (i/2)\delta(t - t')$. The fluctuations in this measurement are characterized by the spectrum of the detection noise (including shot noise) normalized by the total spin noise power, which is called imprecision and denoted $S_{imp}$, and the spectrum of the generalized measurement backaction force acting on the spin, $S_{BA}$. The Heisenberg uncertainty principle constrains the imprecision and the

backaction spectra as $\sqrt{S_{imp} S_{BA}} \geq \hbar/2$ (the spectra are two-sided; see SI Sec. C and ref. 24).

When the probing light is far detuned from optical transitions in the ensemble, the total spin couples to the probe via the position-measurement Hamiltonian $\hat{H}_{int} = -2\hbar\sqrt{\Gamma}\,\hat{X}_L\hat{X}_S$, and modifies the probe variables according to the input–output relations[16,38]

$$\hat{P}_L^{out}(t) = \hat{P}_L^{in}(t) + \sqrt{\Gamma}\,\hat{X}_S(t), \quad \hat{X}_L^{out}(t) = \hat{X}_L^{in}(t), \quad (1)$$

where $\Gamma \propto \Phi N/(A\Delta^2)$ is the measurement rate expressed as a function of the photon flux $\Phi$, the number of atoms $N$, the cross section area of the beam $A$, and the probe detuning from the optical transition $\Delta$[34]. The measurement rate determines the imprecision $\hat{z} = \hat{P}_L^{in}/\sqrt{\Gamma}$, and the quantum backaction force $\hat{F}_{QBA} = 2\hbar\sqrt{\Gamma}\hat{X}_L^{in}$. Ideally, the product of their spectral densities saturates the Heisenberg uncertainty relation, $\sqrt{S_{imp}S_{BA}} = \hbar/2$. When the detection efficiency is below unity, or there are classical noises in the experiment, the backaction-imprecision product is always higher.

The response of the spin to the measurement backaction is described by the Fourier-domain susceptibility $\chi[\Omega] = \Omega_S/(\Omega_S^2 - \Omega^2 - i\Omega\gamma_0)$, where $\Omega_S$ is the resonance Larmor frequency and $\gamma_0$ is the intrinsic decay rate. The response induces correlations between $\hat{X}_L^{out}$ and $\hat{P}_L^{out}$ that can be observed by detecting intermediate quadratures of light, $\hat{Q}_L^\phi = \sin(\phi)\hat{X}_L^{out} + \cos(\phi)\hat{P}_L^{out}$. The spectra of those quadratures, detected by a homodyne detector with efficiency $\eta$, are given by

$$S_\phi[\Omega] = 1/4 + (\eta\Gamma/2)\,\mathrm{Re}(\chi[\Omega])\sin(2\phi) + \eta\Gamma(\Gamma + \gamma_{th})|\chi[\Omega]|^2\cos(\phi)^2, \quad (2)$$

where $\gamma_{th} = (2\,n_{th} + 1)\gamma_0$ is the thermal decoherence rate, and $n_{th}$ is the temperature of the intrinsic bath expressed via the number of excitations (see SI Sec. B3). The term $\propto \cos(\phi)^2$ is due to the spin oscillator motion, and the term $\propto \sin(2\phi)$ is due to the cross-correlation between $\hat{X}_S$ and $\hat{X}_L^{out}$.

Negative cross-correlation can squeeze $S_\phi[\Omega]$ below the vacuum level of 1/4, with the maximum squeezing being at frequencies around $\Omega_S$, where the measurement backaction is resonantly enhanced. The bandwidth around the resonance in which the squeezing is produced is proportional to the measurement rate. As the measurement rate reaches the oscillation frequency, the squeezing band extends to dc.

While the single-oscillator model of the collective spin is sufficient for many purposes, the actual spin has multiple modes, whose annihilation operators can be introduced using the multilevel Holstein–Primakoff approximation[39] as

$$\hat{b}_m = \frac{1}{\sqrt{\Delta N_m}}\sum_{j=1}^N |m+1\rangle_j\langle m|_j. \quad (3)$$

Here, $j = -F, \ldots, F-1$, where $F$ is the ground-state angular momentum number of the atomic species, $m$ is the projection quantum number of the single-atom angular momentum on the x axis, $|m+1\rangle_j\langle m|_j$ are the jump operators between the states $|m\rangle_j$ and $|m+1\rangle_j$ of the individual atoms, and $\Delta N_m = N_{m+1} - N_m$ are the differences in the mean numbers of atoms in the corresponding states. The frequencies of the oscillators are the energy differences between $|m\rangle_j$ and $|m+1\rangle_j$, controlled by an external static magnetic field. The oscillator–light interaction is described by the Hamiltonian (as derived in SI Sec. B, see also refs. 16,40,41)

$$\hat{H}_{int} = -2\hbar\sum_{m=-F}^{F-1}\sqrt{\Gamma_m}\Big(\hat{X}_m\hat{X}_L + \zeta_m\hat{P}_m\hat{P}_L\Big), \quad (4)$$

where the quadratures of the modes satisfy $[\hat{X}_m, \hat{P}_m] = i$, $\Gamma_m$ are the measurement rates, and $\zeta_m = \zeta(2m + 1)/7$ determine the strengths of dynamical backaction (which is a coherent feedback from the light, SI

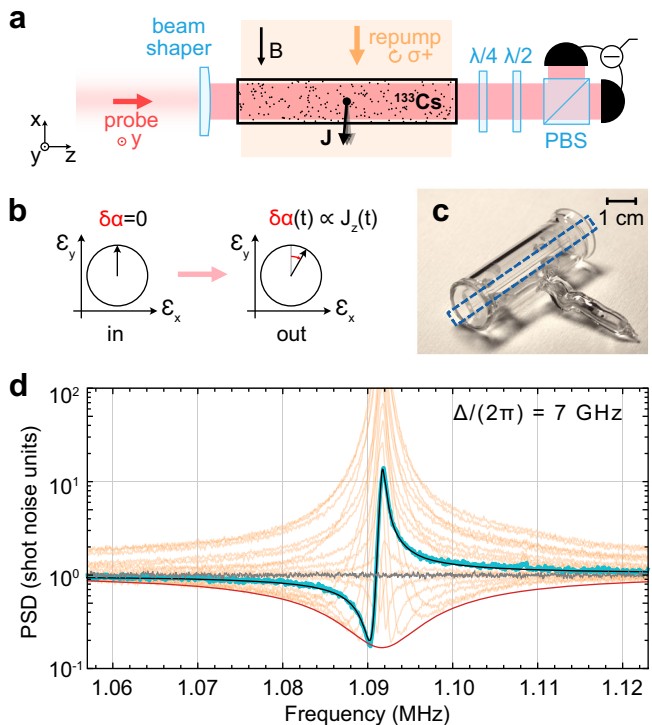

**a** ... beam shaper, B, repump $\circlearrowleft\,\sigma+$, $\lambda/4$ $\lambda/2$, x y z, probe $\odot$ y, $^{133}$Cs, J, PBS

**b** $\delta\alpha = 0$ — $\delta\alpha(t) \propto \hat{J}_z(t)$; $\varepsilon_y$, $\varepsilon_x$, in; $\varepsilon_y$, $\varepsilon_x$, out

**c** 1 cm

**d** $\Delta/(2\pi) = 7$ GHz; PSD (shot noise units) vs Frequency (MHz), 1.06–1.12

**Fig. 1 | Quantum measurements of spin quadratures via Faraday rotation. a** An optical probe spatially shaped in a square tophat beam travels through an atomic ensemble with the total spin $J$ in a magnetic field $B$, and is detected using balanced polarization homodyning. The detected quadrature is selected using the $\lambda/4$ and $\lambda/2$ waveplates. The total spin is oriented by the repump beam traveling along x. PBS: polarization beam splitter. **b** The polarization angle $\delta\alpha$ of the probe as a meter for the spin projection $\hat{J}_z$. **c** A photograph of an anti-spin-relaxation coated cell. The channel with probed atoms is indicated by the blue rectangle. **d** The orange curves show power spectral densities (PSD) of homodyne signals recorded at $\Delta/(2\pi) = 7$ GHz at different quadratures. The trace showing the largest squeezing is highlighted by the blue curve. The black curve is the theoretical prediction based on the global fit, including all quadratures (see the SI). The gray curve is the shot-noise level. The red curve is the theoretical optimum-quadrature squeezing spectrum.

Sec. B and refs. [42–45]). The factor $\zeta$ depends on the atomic level structure and is inversely proportional to the detuning from the optical transition $\Delta$. The effect of the dynamical backaction is optical damping with rates $\gamma_{\mathrm{DBA},m} = 2\zeta_m \Gamma_m$, and an increase in the quantum backaction-imprecision product by an amount proportional to $\zeta^2$ (see the SI), a small number in all our experiments.

The multimode structure of the spin can complicate the interpretation of measurements using the spin as a probe because multiple modes respond to external signals and the measurement backaction in a more complex way than a single mode. Under the conditions of our experiment, however, the response difference is only significant at frequencies close to $\Omega_S$, while far away from $\Omega_S$ the spin always behaves as a single oscillator with $\hat{X}_S = \sum_m \sqrt{\Gamma_m/\Gamma}\,\hat{X}_m$ measured at the total rate $\Gamma = \sum_m \Gamma_m$ and experiencing the decoherence rate $\gamma_{\mathrm{th}} = \sum_m \gamma_{\mathrm{th},m}\Gamma_m/\Gamma$, where $\gamma_{\mathrm{th},m}$ are the individual decoherence rates of the modes.

## Experimental setup

An ensemble of $N \approx 2 \times 10^{10}$ cesium-133 atoms at 52 °C is contained in the $1\,\mathrm{mm} \times 1\,\mathrm{mm} \times 4\,\mathrm{cm}$ channel of a glass chip, shown in Fig. 1c. The channel is coated with paraffin to reduce the spin decoherence from wall collisions[46], and is positioned in a homogeneous magnetic field directed along the $x$ axis (Fig. 1a). The ensemble is continuously probed by a $y-$polarized laser beam propagating in the $z$ direction that has the wavelength 852.3 nm, blue-detuned from the $F = 4 \rightarrow F' = 5$ transition of the D2 line by $\Delta/(2\pi) = 0.7 - 7$ GHz. The ensemble is also continuously repumped using circularly polarized light resonant with the $F = 3 \rightarrow F' = 2, 3, 4$ transitions of the D2 line (see the SI for details). The combination of spontaneous scattering of probe photons and repumping maintains a steady-state distribution of atoms over the magnetic sublevels of the $F = 4$ ground state, which has the macroscopic spin orientation along the magnetic field with polarization $\langle \hat{J}_x \rangle/(NF) \approx 0.78$. The steady-state populations are independent of the probe power in our regime, and correspond to the occupancy of the thermal bath $n_{\mathrm{th}} = 0.9 \pm 0.1$ (see SI Sec. B3, and refs. [47,48]). The resonance frequencies of the oscillators are set by the Larmor frequency and split by 0–40 kHz in different regimes by the quadratic Zeeman and tensor Stark effects. The Larmor frequency can be positive or negative depending on the orientation of the magnetic field, setting the signs of the effective oscillator masses. We work in the negative-

mass configuration[34], but the effects that we observe, in particular the squeezing levels, do not change upon the reversal of the sign of the mass (see the SI). The output light is detected using balanced polarization homodyning, which enables shot-noise-limited detection at frequencies down to 10 kHz.

## High squeezing of light in a tunable narrow frequency band

In Fig. 1d, we present homodyne spectra recorded at the optical detuning $\Delta/(2\pi) = 7$ GHz over a range of detection quadratures $\phi$. In this measurement, dynamical backaction effects are small ($\zeta \approx 0.01$), and the probed spin behaves as a single oscillator subjected to position measurements. The data in Fig. 1d show squeezing down to 7.5 dB, attained by the highlighted blue trace. From a global fit of the spectra at all quadratures using an expression that generalizes Eq. (2) to the case of finite $\zeta$ and extra detection noise (see SI Sec. D), we infer the measurement rate $\Gamma/(2\pi) = 13$ kHz and the quantum cooperativity $\mathcal{C}_q = \Gamma/\gamma_{\mathrm{th}} = 11$.

The measurement rate can be estimated directly from Fig. 1d via the bandwidth $\Delta\Omega$ over which squeezing is present in any of the traces. This bandwidth is the range of optical sideband frequencies for which the nonlinearity of the atomic medium is high enough to modify the variance of the input optical states at the scale of vacuum fluctuations. In the backaction-dominated regime relevant to our experiments, $\Delta\Omega \sim \Gamma$.

The envelope of the traces in Fig. 1d is described by the spectrum given by Eq. (2) minimized over the detection quadrature $\phi$ at each frequency. Neglecting the imaginary part of the response, the optimum-quadrature spectrum is given by

$$S_{\min}[\Omega] = \frac{1}{4} - \frac{\eta}{2}\frac{\Gamma}{\Gamma + \gamma_{\mathrm{th}}} D\!\left(\frac{\Omega - \Omega_S}{\Gamma + \gamma_{\mathrm{th}}}\right), \tag{5}$$

where $D(x) = 1/\left(1 + \sqrt{1 + 4x^2}\right)$. The red curve plotted in Fig. 1d additionally accounts for 0.7 shot noise units of excess $\hat{P}_L$-quadrature noise from the thermal motion of fast-decaying spin modes (see II F). This noise is the main limitation for the backaction-imprecision product in this measurement, which equals $1.5 \times (\hbar/2)$.

Due to the scaling $\Gamma \propto 1/\Delta^2$, higher measurement rates are achievable with the probe laser tuned closer to the atomic transition. In Fig. 2a, we present data obtained at the optical detuning of 3 GHz using 8.4 mW of probe power. In this measurement $\zeta = 0.054$, and the dynamical backaction results in optical damping and hybridization of

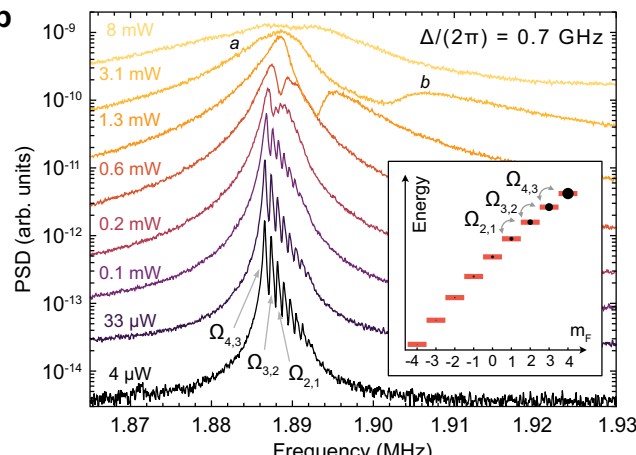

**Fig. 2 | Squeezed light generation by collective spin modes. a** Homodyne signal PSDs at $\Delta/(2\pi) = 3$ GHz and different detection angles $\phi$ indicated in the figure. The points are experimental data. The green and orange traces are obtained close to $\hat{P}_L$ and $\hat{X}_L$, respectively, and the olive, blue and purple—at intermediate quadratures. The gray trace shows the shot-noise level. The black curves are theoretical predictions based on the global fit over 15 quadratures (only 5 are shown, the rest are presented in the SI). The red curve is the optimum-quadrature squeezing spectrum predicted by the single-oscillator model. **b** The spectra of classically driven motion of the collective spin. The eight peaks visible at low probe powers correspond to bare oscillator modes due to the transitions between adjacent $m_F$ levels. Their frequencies are determined by the linear and quadratic Zeeman energies, and magnitudes are determined by the macroscopic populations of the $m_F$ levels as shown in the inset. The spectra at high powers expose the hybridized oscillator modes.

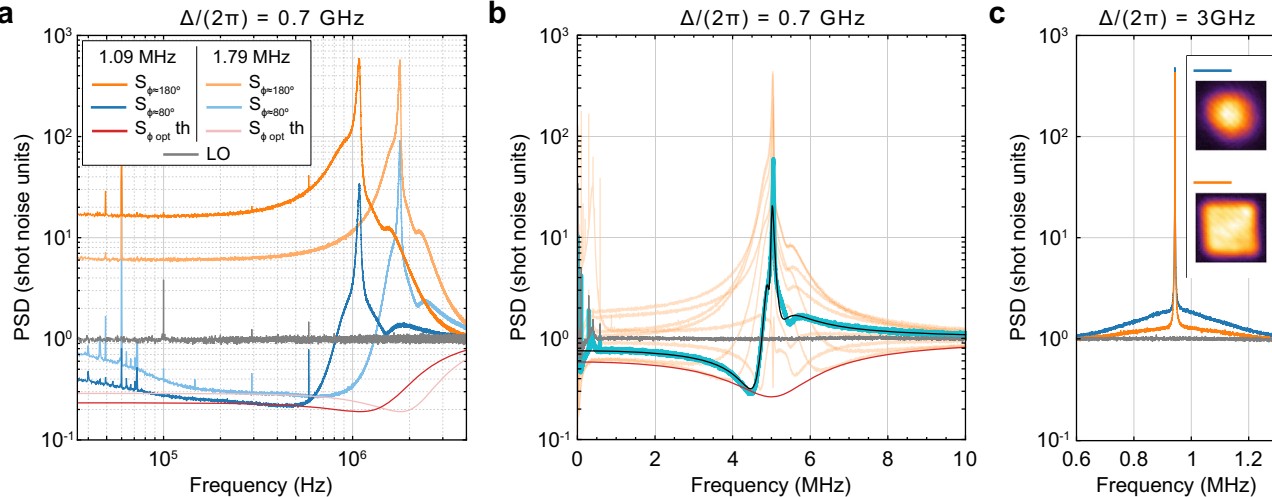

**Fig. 3 | Measurements of spin oscillators outside the rotating-wave regime.**
**a**, **b** Homodyne signal PSDs at $\Delta/(2\pi) = 0.7$ GHz. The gray curves show the experimental shot-noise levels, and the red curves are the theoretical optimum-quadrature squeezing spectra derived from Eq. (2). **a** Spectra for $|\Omega_S|/(2\pi) = 1.09$ MHz and 1.79 MHz. The orange and blue curves are measurements with the quadrature angle set to $\hat{P}_L$ ($\phi \approx 180°$) and close to $\hat{X}_L$ ($\phi \approx 80°$), respectively. LO: local oscillator, th: theoretical. **b** Orange curves show homodyne spectra recorded at

$|\Omega_S|/(2\pi) = 5$ MHz and at different quadratures $\phi$. The trace with the largest squeezing is highlighted by the blue curve. The black curve is the theoretical prediction based on the global fit including all quadratures (see the SI). **c** The spectra taken at the $\hat{P}_L$ quadrature when the probe beam is Gaussian (blue curve) and tophat (orange curve). The gray curve is the shot noise. The inset shows the beam intensity distributions over the 1 mm × 1 mm channel cross section recorded without the cell.

the oscillator modes (SI Sec. B), as well as optical squeezing in the $\hat{X}_L$-quadrature (see the green trace in Fig. 2a). Since the thermal decoherence of the oscillators is due to baths at a temperature close to zero, the optical damping improves the maximum magnitude of squeezing by about 0.5 dB. The minimum noise shown by the blue trace in Fig. 2a is $8.5^{+0.1}_{-0.1}$ dB below the shot-noise level. The overall detection efficiency of our setup is $\eta = (91 \pm 3)$%, and the transmission loss at the exit window of the cell is 1.6%, which means that the magnitude of the squeezing at the exit of the cell is $11.5^{+2.5}_{-1.5}$ dB. The backaction-imprecision product in this measurement is $1.9 \times (\hbar/2)$, which is higher than in the measurement at 7 GHz detuning due to the higher excess $\hat{P}_L$-quadrature noise (two shot-noise units).

### Hybridization of spin modes via coupling to light
The experimental spectra in Fig. 2a can be understood as arising from the coupled dynamics of two nearly degenerate bright modes of the spin, which we refer to as modes $a$ and $b$. To extract their effective parameters, we globally fit the set of spectra recorded over an extended range of quadrature angles (SI Sec. D). We find the total measurement rate $\Gamma/(2\pi) = 52$ kHz, the individual quantum cooperativities $\mathcal{C}_q^a = 12$ and $\mathcal{C}_q^b = 4$, and the total cooperativity $\mathcal{C}_q = 15$. The lower envelope of the experimental traces is in agreement with the optimum-quadrature spectrum predicted by the single-oscillator model using the same $\Gamma$ and $\mathcal{C}_q$.

The bright modes $a$ and $b$ emerge due to the coupling of the individual spin oscillators via the common reservoir of the probe optical modes with coupling rates proportional to $\zeta_m$ and $\Gamma_m$. This coupling approximately scales with the laser detuning as $1/\Delta^3$ at fixed probe power. To illustrate the effect of intermodal coupling, we reduce the laser detuning to 0.7 GHz, where $\zeta = 0.18$. We excite the oscillators with classical white magnetic field noise, and record their driven motion by measuring the $P$ quadrature of the output light. The spectra at different probe powers are shown in Fig. 2b. At the lowest power, the eight bare spin oscillators due to the transitions between adjacent $m_F$ levels are individually resolved. As the probe power is increased, the resonances first merge in two (the $a$ and $b$ modes) and then into three peaks. The macroscopic occupancies of different $m_F$ levels in the

atomic ensemble remain the same at all powers, as is separately checked via magneto-optical resonance signal[47], which means that the change in the output spectrum is only due to the coupled dynamics of the collective oscillators.

### Measurements of the spin at the rate of its precession
At the detuning of 0.7 GHz from the optical transition, the measurement rate of the spin motion can be as high as the oscillation frequency, and the quantum measurement backaction manifests itself via broadband squeezing of light. While the coupling between individual spin oscillators is pronounced at this detuning, at frequencies much lower than $\Omega_S$ the spin behaves as a single oscillator. In Fig. 3a, we present spectra recorded using 12.8 mW of optical probe power at two resonance frequencies, 1.09 MHz and 1.79 MHz, in which the bandwidth of low-frequency squeezing extends down to 30 kHz. The minimum noise levels of the homodyne signals (6.5 dB below the shot noise for the 1.09 MHz data) are consistent with the quantum cooperativity $\mathcal{C}_q = 8$. The measurement rate can be estimated from the signal-to-shot-noise ratio on the $P$ quadrature in Fig. 3a using Eq. (2) in the limit $\Gamma \gg \gamma_{th}$,

$$S_{\phi=0}[\Omega = 0] = 1/4 + \eta(\Gamma/\Omega_S)^2, \qquad (6)$$

which yields $\Gamma/(2\pi) \approx 2$ MHz, a value higher than the resonance frequencies. To further corroborate the measurement rate, we perform a quadrature sweep with the resonance frequency set to 5 MHz and using 10.2 mW of probe power (Fig. 3b). From fitting this data, we find $\Gamma/(2\pi) = 1.77$ MHz, which is consistent within ten percent with the previous estimate corrected for the difference in the probe powers. Theoretically, the optimum-quadrature noise levels should saturate as the Fourier frequency approaches zero, to a value around 0.22 shot-noise units for the 1.09 MHz data in Fig. 3a. In the experiment, the noise levels increase at low frequencies due to excess noise from the atomic ensemble.

The backaction-imprecision product for the measurements in Fig. 3a is below $1.2 \times (\hbar/2)$ at frequencies higher than 100 kHz. This value is closer to saturating the Heisenberg uncertainty relation than

the values in the slow-measurement experiments, because the fast-decaying modes are in the backaction-dominated regime, and do not contribute excess thermal noise. The limiting factors for the product in this case are the dynamical backaction and detection inefficiency.

### Fast-decaying spin modes

In addition to the collective oscillators described by the annihilation operators from Eq. (3), in which all atoms contribute equally, there are other modes of the spin in our system[40,49]. The resonance frequencies of these modes coincide with $\Omega_S$, but their decay rates are limited by the rate of atoms flying through the probe field ($\gamma_{0,\text{flight}}/(2\pi) \approx 300$ kHz) rather than collisions with the walls and other atoms ($\gamma_{0,\text{coll}}/(2\pi) \approx 200$ Hz). The annihilation operators of these modes are

$$\hat{b}'_m = \frac{1}{\sqrt{\Delta N_m \left\langle \Delta g(t)^2 \right\rangle_c}} \sum_{j=1}^{N} \Delta g_j(t) |m\rangle_j \langle m+1|_j, \tag{7}$$

where $g_j(t)$ are the coupling rates between the optical probe and the individual atoms (SI Sec. B) and $\langle \Delta g^2 \rangle_c$ is the squared deviation of the coupling from the mean averaged over classical trajectories, assumed to be the same for all atoms. The measurement rate of the fast-decaying modes is $\propto \langle \Delta g^2 \rangle_c$, while the measurement rate of the slow-decaying modes is $\propto \langle g \rangle_c^2$.

An enabling feature of our experiment is the high 3D uniformity of the optical probe field, achieved using a tophat beam configuration, which reduces $\langle \Delta g^2 \rangle_c$ and thus the readout of the fast-decaying modes. In Fig. 3c, we compare the spectra recorded at the $\hat{P}_L$-quadrature using a tophat and a wide Gaussian probe beam (the exact beam profiles are given in SI Sec. A) with equal optical powers in the slow-measurement regime. The thermal noise contributed by the fast-decaying modes is reduced from 1 to 0.3 shot-noise units on resonance upon switching from the Gaussian to the tophat probe. The absolute non-uniformity of the coupling[50,51] for the tophat beam is estimated to be $\langle \Delta g^2 \rangle_c / \langle g \rangle_c^2 = 0.6$ based on the camera imaging.

## Discussion

Continuous measurements that combine high measurement rate, quantum cooperativity, and detection efficiency can be used for single-shot generation of spin-squeezed states and quantum state tomography[52]. The entanglement link between the material spin and traveling light entailed by the squeezing enables quantum-coherent coupling of spins with other material systems[16,35]. While the backaction-imprecision product in all our measurements is already within a factor of two from the Heisenberg bound, it can be further improved by optimizing the probe power for measurements of the $\hat{P}_L$-quadrature. Our measurements were optimized for quadratures intermediate between $\hat{X}_L$ and $\hat{P}_L$ (i.e., for "variational" readout[25]), which can yield superior results[53] in quantum sensing and control.

This work also establishes room-temperature atomic spin oscillators as a practical platform for engineering quantum light with high levels of squeezing, which is a basic resource for interferometric sensing and optical quantum information processing[22]. The highest demonstrated squeezing, reaching 8.5 dB at the detection, is narrow-band, but its frequency can be tuned by the magnetic field without degrading the level within the range of ~0.8–5 MHz in our experiments.

## Methods

Extended details of the measurement setups, additional experimental data, and the theoretical description of the spin–light interaction are presented in Supplementary Information.

## Data availability

The data presented in the plots, together with the analysis scripts, are available at Zenodo (https://doi.org/10.5281/zenodo.10927399).

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

## Acknowledgements

The authors thank Michał Parniak, Jörg Müller, Rebecca Schmieg, and Ivan Galinskiy for general help and useful discussions, and also Chao Meng for the discussions of spin squeezing via fast position measurements. This work was supported by the European Research Council (ERC) under the Horizon 2020 (grant agreement No. 787520), VILLUM FONDEN under a Villum Investigator Grant no. 25880, and Novo Nordisk Foundation (grant NNF20OC0059939 'Quantum for Life'). S.F. acknowledges funding from the European Union's Horizon 2020 research program under the Marie Sklodowska-Curie grant agreement No. 847523 "INTERACTIONS".

## Author contributions

C.B. and S.A.F. performed the experiments, with help from C.Ø. C.B. led the construction of the experimental setup. S.A.F. developed the theory, with help from C.B., E.Z., and E.S.P. M.V.B. fabricated the vapor cell. E.S.P. supervised the project. All authors contributed to the planning of the experiment and writing the paper.

## Competing interests

The authors declare no competing interests.
