## [Peer Review File · Nature Communications]

Squeezed light from an oscillator measured at the rate of oscillationReviewer #1 (Remarks to the Author):

The authors report high degrees of light squeezing in an atomic vapor cell, more than 10dB. The result itself is very impressive, with the squeezing level higher than any previous experiments using atomic medium (as far as I know), approaching that achieved in traditional setups using crystals. The authors presented both experiment and theoretical results, which are in good agreement. The mechanism for light squeezing is a nice extension of what the same group used before in a two-cell experiment (Optics Express 2009). I believe this work is of high quality, and also of high impact in general given the strong interest in generating squeezed light for various applications of quantum information science and quantum metrology. I therefore would recommend publication in Nature Communications, if the authors would consider the following comments to improve the clarity of the paper for the general audience.

1. "Measurement rate" is a key terminology used throughout the paper, however it is unclear what exactly this means, especially given that all the laser fields used here are continuous (nothing is pulsed). Of course, the oscillator(s) has its oscillation frequency (Larmor frequency), but when viewed using the quantization axis along the magnetic field, there is just Zeeman splitting, so nothing in the system is being pulsed. I would think the measurement is continuous. It seems like that the "measurement rate" is associated with the atom-light coupling strength.

2. In all figures, why does the red curve deviate so much from experiments?

3. In line 106, what does the "two-sided spectra" mean? Does it mean the Stokes and Anti-Stokes field's frequencies lie at the two sides of the input probe's frequency?

4. "Multi-mode" structure is mentioned many times, such as in line 143, but this has not been defined. Does it refer to the multiple oscillators with different frequencies? Also, is it an important (useful) feature that they all have different frequencies, or an unwanted feature due to quadratic Zeeman and tensor light shift?

5. In Fig. 2b, when the laser power increases, the spectrum starts to have extra peaks on the right hand side (the 3.1mW, 1.3mW data), which is hard to understand if one just thinks about power broadening. Some physics explanation is needed. Also, in this figure, the y-axis is a.u., then is the base for all curves according to the scale? i.e., is the base at higher power many orders of magnitude higher than those for lower laser power?

6. It is mentioned that the population distribution does not change for a wide variety of probe laser powers. This is quite counter-intuitive, given that the probe is near resonance and its power is comparable to the repump. Please explain.

7. Tophat beam is used here to reduce the readout of the fast decaying modes. However, in a recent paper on spin squeezing [PHYSICAL REVIEW LETTERS 130, 203602 (2023)] by the same group, tophat beam is not used. Since tophat beam would usually increase the k-vector variation of the probe (due to the uncertainty relation of position and momentum for light), how does this affect the experiment? I assume it would degrade spin squeezing, but maybe for light squeezing this is not so critical?

8. "Optical damping" in the supplementary material is mentioned. Please explain what this means. Is this just power broadening?

9. How does the squeezing level depend on the polarization of the atoms? Do you get more squeezing if the polarization approaches 1 (it is 0.78 in the paper)? Do you still have light squeezing if the repump laser is turned off (thermal spin state)?

10. What's the main difference of the mechanism for light squeezing used here, compared to the one used in the two-cell experiment (where the squeezing quadrature is fixed at the z-direction), and compared to light squeezing based on (degenerate-)four-wave mixing process (also called polarization-self-rotation) where the squeezing quadrature varies with the coupling strength too?

Some typos :

1. In Fig. 1, the polarization of the probe is indicated by a little red vertical arrow with the letter "y" next to it. I believe the red arrow points to a wrong direction if it is y-polarization.
2. Line 110, $\sin(2\theta)$ should be $\sin(2\phi)$.

Reviewer #2 (Remarks to the Author):

The authors present experimental results in which squeezed light is obtained in a "measurement + feedback" type scheme where the "meter" is a light beam and the measured system is an ensemble of Cesium atoms in a cell at room temperature.

The polarization degrees of freedom of the light are represented by the Stokes vector and the internal degrees of freedom of the atoms by a collective spin which, in the presence of a magnetic field, behaves like an oscillator.

The authors identify two distinct regimes in which a significant reduction of fluctuations below the standard quantum limit (squeezing) is observed in a quadrature of the meter at the output of the cell, after interaction with the atoms. In the first regime, which corresponds to a measurement rate smaller than the atomic Larmor frequency, they observe a large amount of light squeezing for frequencies close to resonance. In the second regime, in which the measurement rate is of the order of the Larmor frequency, light squeezing is observed over a wide band of low frequencies.

The results presented in the paper are important both because they identify room-temperature atomic vapors as competitive systems for squeezed light production, and because they represent a significant progress in understanding and using room-temperature vapors as atom-light quantum interfaces. The latter point is remarkable considering the complexity of the system. A first theoretical complication is related to the presence of more than two levels in the ground state of the atoms, which complicates the atom-light interaction and forces the introduction of 2F oscillators (instead of just one) to describe the internal degrees of freedom of the atomic collective spin. A second complication is due to the presence of several spatial modes in the vapor and to the inhomogeneities and fluctuating character of the interaction between light and atoms, caused in particular by the transit of atoms through the light beam. This requires the introduction of additional, strongly damped, oscillators in the theoretical description.

Given the importance of the results, I recommend publication of the article in Nature Communications. Below are some comments/questions that might improve the readability of the article.

1) Several important quantities, such as the "two-sided spectral densities" S_{imp} , S_{phi} and the feedback S_{BA} , are not defined in the text.

2) Equation (4) is important as it describes the deviation of the interaction from the simple form $X_L X_S$ due to the presence of various levels in the ground atomic state. I think the authors should explain where and under what assumptions this formula was derived, pointing to the important equations and sections in the Supplementary Information or other references.

3) Several times in the article the notion of thermal occupation of oscillator levels appears. I do not quite understand what this thermal distribution corresponds to, how it is obtained, and what the

corresponding temperature depends on. If the answer is to be found in the supplementary material, the authors should indicate a precise reference to the section and equations.

Reviewer #3 (Remarks to the Author):

In their manuscript, Baerentsen et. al. look at the dynamics of an atomic spin probed by light, and investigate how the Faraday interaction results in squeezing of the effective spin. This is done for different regimes, characterised by the ratio between the measurement rate and the spin oscillator frequency.

The study is potentially interesting, but the text is extremely hard to follow even for someone with a bit of experience in QND-based squeezing, and light-matter Faraday interaction. The language used is very technical and it requires knowledge of previous experiments from the group. Moreover, it is often lacking a clear and intuitive explanation of the physics behind the effects investigated.

I list below a number of points that I found particularly confusing, and that should be better addressed or explained in the text.

- in the introduction, why the authors mention POVMs rather than QND measurements? the link between these two concepts should be clarified to the reader.
- when talking about position measurement of the oscillator, it should be mentioned in which frame this is performed. I guess the lab frame.
- "When the rate is slower than the oscillation, measurements with the meter in the vacuum input state project the oscillator on coherent states." is this simply because when measuring the position of the oscillator in the lab frame for long time results in an averaging over a "rotating quadrature"? So there is a continuous squeezing and anti-squeezing as the system evolves?
- how does this measurement rate faster than the oscillation compare to the stroboscopic measurement approach that was also developed by the same group? If the interaction is weak, which results in a slow measurement rate, one can always decompose the measurement into shorter and stroboscopic pulses. Is this correct? What would be the difference (e.g. in terms of bandwidth)?
- what does it mean "the rate of position measurement affects the output state of the meter field"? Is it really the rate (i.e. the "speed") that affects the field? Or the authors are just referring to the measurement back-action?
- when saying "In the slow measurement regime, the correlations and the associated squeezing exist in a narrow frequency band near the resonance, and have a strong frequency dependence due to the time-averaged response of the oscillator to the measurement backaction", what does it mean the strong frequency dependence? The sentence already says squeezing exist in a narrow frequency band, so is this a redundant statement or the authors mean something else that is not explained?
- The author should give a clear physical and intuitive picture of why for measurement rates faster than the oscillation the squeezing of the field is broadband.
- Is it true that in the slow measurement regime the oscillator is projected into a coherent state, but the light field get squeezed? This is not explained clearly.
- "quantum fluctuations of ponderomotive torque" is not explained.

-- "The Heisenberg uncertainty principle constrains the two-sided spectral densities of the imprecision in ...", it is not clear. Someone not familiar with the connection between spectral density and oscillator dynamics is required to read extra literature and SI, but this measurements are crucial to perform quadrature measurements and should be explained in the text for the article to be self-contained.

-- How is Eq.2 derived? Details should we put in the SI.

-- Is the term " $P_m \cdot PL$ " in Eq.4 coming from tensor polarisability? The term "dynamical backaction." is not explained, and is not clear what it means. Missing citations to previous works of the group where this Hamiltonian is derived in detail.

-- what does it mean "The multimode structure can affect the response of the spin to the measurement backaction at frequencies close to Ω_S "? Affect in which way? Why?

-- when discussing the population of the F sub-levels, can the authors give quantitative numbers? What fraction of atoms occupy each of the m_F levels?

-- "We work in the negative-mass configuration", why? what would be different with a positive mass oscillator (ie opposite field orientation)?

-- For all figures, it is missing the experimental sequence used to take the measurements. Example: Fig.1d, is the result of only one light pulse continuously measured? How long? Or there is one state-preparation pulse, followed by a read-out pulse? Is there any operation done between consecutive shots to "reset" the system? Or just waiting?
Similar description of the sequences are needed for each figure.

-- "From a global fit of the spectra at all quadratures", with what function? Eq.2?

-- The emergence of bright modes is discussed on the right of Eq.5 only very qualitatively. This part should be better explained and made more rigorous. The merging of the eight peaks into two and three broader peaks is neither explained nor modelled, and an intuitive picture of the physical process is completely missing.

-- As a curiosity, how is it achieved shot-noise-limited readout with a power of $\sim 12\text{mW}$? What are the details of the detector, and of the readout setup? What is the minimum field quadrature noise that could be measured (ie maximum squeezing)?

-- Details on the derivation of Eq.6 are missing.

-- I did not find clear how to understand the peaks in Fig.2b, and the energy spectrum on the inset. Are these the $2F$ modes coming from performing the HP transformation of each angular momentum?

-- what is the physics behind these transitions between m_F levels? How does it happen? Why classically driven motion does not simply result in a coherent state of the modes defined through the HP transformation?

-- The coupling between the modes, as a function of the magnetic field, should be better explained and investigated. How the coupling mentioned in the text above Eq.6 depends on the detuning? It is not explained the physical mechanism behind this.

We thank all the reviewers for their comments.

Reviewer #1:

The authors report high degrees of light squeezing in an atomic vapor cell, more than 10dB. The result itself is very impressive, with the squeezing level higher than any previous experiments using atomic medium (as far as I know), approaching that achieved in traditional setups using crystals. The authors presented both experiment and theoretical results, which are in good agreement. The mechanism for light squeezing is a nice extension of what the same group used before in a two-cell experiment (Optics Express 2009). I believe this work is of high quality, and also of high impact in general given the strong interest in generating squeezed light for various applications of quantum information science and quantum metrology. I therefore would recommend publication in Nature Communications, if the authors would consider the following comments to improve the clarity of the paper for the general audience.

We thank the referee for the thorough and comprehensive review, and we very much appreciate his/her feedback. Admittedly, there is a terminology gap that exists between different communities working with quantum measurements, which we have tried to bridge in the revised manuscript by adding more comprehensive and self-contained introductions of the concepts we operate with.

1. “Measurement rate” is a key terminology used throughout the paper, however it is unclear what exactly this means, especially given that all the laser fields used here are continuous (nothing is pulsed). Of course, the oscillator(s) has its oscillation frequency (Larmor frequency), but when viewed using the quantization axis along the magnetic field, there is just Zeeman splitting, so nothing in the system is being pulsed. I would think the measurement is continuous. It seems like that the “measurement rate” is associated with the atom-light coupling strength.

All of this is correct – the measurement is continuous, nothing is pulsed, and the quantity that we call “measurement rate” characterizes the steady-state atom-light coupling.

The thought model motivating this terminology is the following. A monitored system (the spin of the ensemble in our case) interacts with an infinite sequence of bosonic probes (the temporal modes of the light field) one after another. The interactions with individual probes are weak and quick, so that the coarse-grained time dynamics are continuous. The probes after the interaction are projectively measured via homodyne detection, thus providing a record of indirect measurements of the spin.

What is called the rate of the indirect measurement in this model is a limit of the frequency with which the probes are incident normalized by their individual coupling strength. It quantifies how quick information about the system can be extracted from the probes assuming their detection is optimum in some sense.

This model is explained, for example, in the review A. A. Clerk, M. H. Devoret, S. M. Girvin, F. Marquardt, and R. J. Schoelkopf “Introduction to Quantum Noise, Measurement and Amplification.” Rev. Mod. Phys. 82, 1155–1208 (2010) for the case of measurements on superconducting qubits (and therefore with no oscillation frequency). The notion of the measurement rate is widely used in works that perform continuous measurements on material harmonic oscillators, such as optomechanical cavities and

levitated nanoparticles (Ref. [20, 21, 29-32] in our list of literature use this terminology), and was used for atomic ensembles before (Ref. [16, 34]).

In the revised manuscript, we elaborate just below Eq. 1 how the definition of the measurement rate relates to the experimental parameters. We write that $\Gamma \propto \Phi N / (A \Delta^2)$, where N is the number of atoms, A is the cross section area of the beam, and Δ is the probe detuning [34], and that operationally Γ defines the ratio of the contribution of the spin noise to the shot noise in the detected optical noise power. Γ is proportional to the optical depth N/A which is a characteristic figure of merit for quantum coupling between light and an atomic ensemble (Ref. [4, 16, 34]).

2. In all figures, why does the red curve deviate so much from experiments?

There are three aspects that need to be mentioned to answer this question.

1. First, the red curve is an envelope over the quadrature variation, which means that it should agree with each of the measured traces only at one particular trace-dependent frequency. When we show many traces taken at different quadratures (Fig 1d of the main text), the envelope closely follows the overlapped experimental data.

2. Second, the red curve is calculated neglecting the intrinsic decoherence of the spin. It gives an excellent approximation away from the Larmor frequency, it is not a good approximation around the Larmor frequency.

3. Third, specific to the data in Fig 3a at low frequencies. There is some classical noise (likely, at least in part, originating from the probe laser) that starts being visible at frequencies below 100 kHz.

3. In line 106, what does the “two-sided spectra” mean? Does it mean the Stokes and Anti-Stokes field’s frequencies lie at the two sides of the input probe’s frequency?

This refers to how the spectra of the noise processes are treated, and all frequencies in question are the “low” frequencies of the spin dynamics, and not the “high” optical frequencies. A two-sided spectrum, S_{xx} , is formally defined for frequencies that span the whole real axis, and is normalized so that

$$\int_{-\infty}^{\infty} S_{xx}[\omega] \frac{d\omega}{2\pi} = \langle x^2 \rangle,$$

while and the corresponding one-sided spectrum, S_x , is defined for frequencies $\omega > 0$, and normalized so that

$$\int_0^{\infty} S_x[\omega] \frac{d\omega}{2\pi} = \langle x^2 \rangle.$$

Therefore, the relation between the spectra is $S_x[\omega] = 2S_{xx}[\omega]$ for $\omega > 0$ and $S_x[0] = S_{xx}[0]$.

The one-sided convention for the spectra prevails in the Gravitational Wave Detection community that contributed a lot of fundamental research on continuous measurements, so we felt it would be appropriate to make a remark that we are not following this convention.

If we used one-sided spectra, the value of the backaction-imprecision product limited by the Heisenberg uncertainty principle would be \hbar , as compared to $\hbar/2$ in our manuscript. Otherwise, the normalization convention only has an effect for the theoretical calculations, because the data in all the experimental

plots we provided are relative to the shot noise, making the presentation independent of the normalization convention.

We added a short version of the above explanation to Section C in the supplementary information, where we introduce and talk about the spectral densities, and a reference to Section C where we mention two-sided spectra in the main text.

4. “Multi-mode” structure is mentioned many times, such as in line 143, but this has not been defined. Does it refer to the multiple oscillators with different frequencies? Also, is it an important (useful) feature that they all have different frequencies, or an unwanted feature due to quadratic Zeeman and tensor light shift?

Indeed, the “multi-mode” structure in the manuscript refers to multiple oscillators within the collective spin, which have different frequencies split by the quadratic Zeeman and tensor Stark effects. Importantly, in addition to different frequencies, the oscillators have somewhat different couplings to the light field, because their ζ -coefficients are different. (ζ_m are defined after Eq. 4 and given in terms of the atomic polarizability parameters a_1 and a_2 after Eq. SI B 11).

This effect of having multiple oscillator modes is neither beneficial nor detrimental for the observed level of squeezing. The dynamics of the multiple (coupled) modes, however, are much more complicated than the dynamics of a single mode would be, which affects our ability to theoretically model the output noise spectra. We believe that it is the imperfection of the knowledge of the multi-mode dynamics that limits how well our theory models fit the experimental data.

In the revised manuscript, we explicitly state that the oscillators and the modes refer to the same thing. In the same paragraph that the Reviewer mentions, we also write that for us the main effect of multiple modes is to complicate the interpretation of the measurement spectra.

5. In Fig. 2b, when the laser power increases, the spectrum starts to have extra peaks on the right hand side (the 3.1mW, 1.3mW data), which is hard to understand if one just thinks about power broadening. Some physics explanation is needed. Also, in this figure, the y-axis is a.u., then is the base for all curves according to the scale? i.e., is the base at higher power many orders of magnitude higher than those for lower laser power?

The spin signal peaks visible at high optical powers, including the one on the right (which is also present in the 8 mW trace, but outside of the shown span of data, at a higher frequency), are products of hybridization between different spin oscillator modes due to the dissipative coupling via the optical field. Also see Resp. Fig. 1 in the response to Reviewer 3.

To explain the physics behind this effect, it is useful to consider first only one oscillator mode coupled to a traveling optical field. Due to the tensor polarizability, contributing the $P_L P_S$ term to the Hamiltonian, part of the signal picked up by the field acts back on the spin oscillator in the form of a force, introducing an effective feedback loop. This “feedback” can amplify or dampen the spin motion. The damping rate that it produces is what we call **optical damping**, $\gamma_{opt} = 2\zeta\Gamma$. Another term used in the literature to denote this effect is dynamical backaction, which is how it is referred to in the SI.

In the revised Supplementary Information, Sec. B, we have a dedicated subsection that explains the terminology related to the dynamical backaction and optical damping. We refer the reader to this subsection when introducing the optical damping in the main text.

When there are multiple spin oscillators interacting with light in presence of the described feedback mechanism (which is always present unless all ζ - coefficients are zero), this same mechanism also couples different oscillators. This happens because the feedback force driving each individual oscillator has contribution from all other oscillators. The coupling terms are given explicitly in Eq. SI B 23-24 (Supplementary section B). There, n and $m = -F, \dots, F - 1$ index the oscillator modes, and the dynamical equations for the quadratures of each mode, dX_m/dt and dP_m/dt , contain contributions from all other modes n .

Because of the optics-mediated coupling, bare spin oscillators hybridize. This happens only at optical powers sufficiently high for the coupling to be larger than a) the intrinsic damping and b) the mode splitting due to quadratic Zeeman effect. The emergence of the hybridization pattern and its evolution versus optical power is what is shown in Fig. 2b. The fact that the hybridized modes consist of one or two peaks that stay almost perfectly at the Larmor frequency, and one peak shifts to the higher frequency seems to be coincidental – we could qualitatively reproduce it in a first-principle calculation, but we can't offer much intuition why it happens.

At present, our attempts to model this effect from first principles have been of limited success – while qualitatively we can reproduce the emergence of the hybridized peaks, some details of the spectra do not completely agree with the measured data, which is why for the presentation in the manuscript we choose to fit the data with a model having some phenomenological parameters.

The arbitrary units in Fig 2b mean that there is an arbitrary scaling factor applied to the data – the same factor for all traces. So yes, the data taken at high optical powers display photocurrent spectral densities many orders of magnitude higher than the data taken at low powers.

6. It is mentioned that the population distribution does not change for a wide variety of probe laser powers. This is quite counter-intuitive, given that the probe is near resonance and its power is comparable to the repump. Please explain.

Although the probe and the repump lasers have comparable optical powers, the repump is resonant with $F=3 \rightarrow F'=2, 3, 4$ transitions within the Doppler linewidth, while the probe is detuned from $F=4 \rightarrow F'=3, 4, 5$ transitions by ≥ 2 FWHM Doppler linewidths. Therefore, the repump keeps the occupation of $F=3$ manifold negligible, doing its normal job.

The macroscopic steady-state distribution of atoms over the magnetic sublevels of $F=4$ manifold is determined by the rates at which the individual atoms can jump between the magnetic sublevels of $F=4$ with different m_F s. An atom starting in a certain m_F sublevel of $F=4$ can make a jump to another m_F sublevel of $F=4$ via one of the two processes:

- a) It can absorb a photon from the probe laser, then spontaneously emit a photon to end up in a different m_F state of $F=4$.
- b) It can absorb a photon from the probe laser, then spontaneously emit a photon to end up in $F=3$, then absorb a repump photon, then spontaneously emit again either to return to $F=4$, or to return to $F=3$ and repeat the last two stages again until the spontaneous emission event brings

the atom to $F=4$. This full sequence still corresponds to one jump when seen from the perspective of atoms performing random walks between the mF sublevels of $F=4$.

The rates of both (a) and (b) processes are proportional to the probe power. Since the steady state distribution over mF s of $F=4$ depends only on the ratio of the jump rates, this distribution is independent of the probe power (of course, down to the point when the optical power is so low that other mechanisms than spontaneous scattering begin dominating the inter- mF relaxation). The absolute rates at which atoms jump, of course, are proportional to the probe power.

Independent of this theoretical argument, this fact that the distribution of atoms over the magnetic sublevels of $F=4$ is independent of the probe power within our experimental uncertainties was checked using the magneto-optical resonance method (described in Julsgaard et al “Characterizing the spin state of an atomic ensemble using the magneto-optical resonance method.” J. Opt. B: Quantum Semiclass. Opt. 6, 5 2003).

Another check that we made is that upon changing the repump power within some range the distribution of the fractions of atoms across the $F=4$ manifold remained unchanged, and what varied was only the number of atoms lost to the $F=3$ manifold. The probe interacts negligibly weakly with atoms in $F=3$, and, therefore, its population only leads to a small reduction of cooperativity, in the first approximation.

7. Tophat beam is used here to reduce the readout of the fast decaying modes. However, in a recent paper on spin squeezing [PHYSICAL REVIEW LETTERS 130, 203602 (2023)] by the same group, tophat beam is not used. Since tophat beam would usually increase the k -vector variation of the probe (due to the uncertainty relation of position and momentum for light), how does this affect the experiment? I assume it would degrade spin squeezing, but maybe for light squeezing this is not so critical?

The degree to which the k -vector varies due to the transverse shape of the probe beam depends on the beam cross-section. In the present experiment the beam cross-section is 1mm by 1mm. The corresponding Rayleigh range is much larger than the cell length, which makes the uncertainty that the Referee mentions negligible.

The use of the top hat beam is critical for our experiment, since it allows to maximize the fraction of atoms that experience multiple interactions with light and to minimize the fraction of atoms which experience that interaction only once. This is illustrated in Fig. 3c where the reduction of the broadband spin noise due to the latter type of interaction due to the top hat probe is clearly seen. Without the top hat beam shape the degree of light squeezing would be substantially lower.

In comparison, the degree of spin squeezing which has been the subject of the PRL 130, 203602 (2023) is less sensitive to the presence of the broadband noise, although it would also be beneficial to use a top hat beam there.

8. “Optical damping” in the supplementary material is mentioned. Please explain what this means. Is this just power broadening?

The meaning of “optical damping” is explained in the answer to question #5. The total damping added to the spin by the probe light is the optical damping (equal to $2\zeta\Gamma$) plus power broadening, which is damping due to the incoherent scattering of photons. Optical damping and power broadening are of

similar magnitudes in our experiments. Both of them are proportional to the probe power, but, at a fixed probe power, they have different scaling with the optical depth along the probe propagation direction – while the optical damping is proportional to the optical depth, power broadening is independent of the optical depth.

In the revised manuscript, we refer to a new SI subsection of section B for the explanation of dynamical backaction and optical damping in particular. There, we also explain how such an optical damping is different from power broadening.

In addition to the above, please see the responses to questions #1 and #5 for more information.

9. How does the squeezing level depend on the polarization of the atoms? Do you get more squeezing if the polarization approaches 1 (it is 0.78 in the paper)? Do you still have light squeezing if the repump laser is turned off (thermal spin state)?

This is an interesting question, because the squeezing level is not linked to the polarization very straightforwardly. If the polarization could be increased from 0.78 to 1 without adding decoherence for the spins (e.g., if the optical depth of the ensemble was infinite), the increase in polarization would, indeed, improve the squeezing level, because the cooperativity would increase proportionally to the polarization, and the effective occupancy of the thermal bath of the spin would reduce from around 1 to 0 quanta. However, in the real world, where polarization is increased by illuminating the ensemble with an additional circularly polarized pump laser (resonant with $D1 F=4 \rightarrow F'=4$ transition in our experiment), the effect of pumping on the squeezing is generally negative. In our investigations so far (which are beyond the scope of the manuscript), the extra decoherence due to the pump light reduced the quantum cooperativity by an amount that was not compensated by the increase in the coupling rate and the reduction in the bath occupancy. Therefore, practically the repump-only configuration seemed to be optimal for the squeezing level.

Answering the second part of the question – no, there is no light squeezing when the repump laser is off and the ensemble is in the thermal state.

10. What's the main difference of the mechanism for light squeezing used here, compared to the one used in the two-cell experiment (where the squeezing quadrature is fixed at the z-direction), and compared to light squeezing based on (degenerate-)four-wave mixing process (also called polarization-self-rotation) where the squeezing quadrature varies with the coupling strength too?

The physical mechanism responsible for light squeezing in our work is the same as in the two-cell experiment by Wasilewski et al. "Generation of two-mode squeezed and entangled light in a single temporal and spatial mode." *Opt. Express*, OE 17, 14444, (2009). Although the two-cell configuration was experimentally more complex, the end effect that the spin ensembles produced on the light was simpler to interpret. This was because in the two-cell experiment the ensembles were prepared with opposite effective masses, therefore many effects of light-spin interaction had opposite signs and cancelled out in the detection. This was particularly useful to cancel the effect of the classical laser noise driving the spin ensemble, but also resulted in the quadrature dependence of the squeezing being simpler.

Polarization self-rotation (PSR) is a different effect. Both PSR and the mechanism in our work rely on circular birefringence induced in the atomic ensemble by vacuum fluctuations of light. However, for PSR

the birefringence is mainly due to differential Stark shifts of the magnetic sublevels, while for our mechanism it is due to the rotation of the total atomic spin. Therefore, for our mechanism it is necessary to induce a macroscopic polarization in the atomic ensemble perpendicular to the direction of propagation of light, while PSR does not require any spin polarization.

Some typos :

1. In Fig.1, the polarization of the probe is indicated by a little red vertical arrow with the letter “y” next to it. I believe the red arrow points to a wrong direction if it is y-polarization.

2. Line 110, $\sin(2\theta)$ should be $\sin(2\pi)$.

We thank the reviewer for making us aware of these typos. They are corrected in the revised manuscript.

Reviewer #2:

The authors present experimental results in which squeezed light is obtained in a "measurement + feedback" type scheme where the "meter" is a light beam and the measured system is an ensemble of Cesium atoms in a cell at room temperature.

The polarization degrees of freedom of the light are represented by the Stokes vector and the internal degrees of freedom of the atoms by a collective spin which, in the presence of a magnetic field, behaves like an oscillator.

The authors identify two distinct regimes in which a significant reduction of fluctuations below the standard quantum limit (squeezing) is observed in a quadrature of the meter at the output of the cell, after interaction with the atoms. In the first regime, which corresponds to a measurement rate smaller than the atomic Larmor frequency, they observe a large amount of light squeezing for frequencies close to resonance. In the second regime, in which the measurement rate is of the order of the Larmor frequency, light squeezing is observed over a wide band of low frequencies.

The results presented in the paper are important both because they identify room-temperature atomic vapors as competitive systems for squeezed light production, and because they represent a significant progress in understanding and using room-temperature vapors as atom-light quantum interfaces. The latter point is remarkable considering the complexity of the system. A first theoretical complication is related to the presence of more than two levels in the ground state of the atoms, which complicates the atom-light interaction and forces the introduction of $2F$ oscillators (instead of just one) to describe the internal degrees of freedom of the atomic collective spin. A second complication is due to the presence of several spatial modes in the vapor and to the inhomogeneities and fluctuating character of the interaction between light and atoms, caused in particular by the transit of atoms through the light beam. This requires the introduction of additional, strongly damped, oscillators in the theoretical description.

Given the importance of the results, I recommend publication of the article in Nature Communications. Below are some comments/questions that might improve the readability of the article.

We are grateful to the Reviewer for the high appraisal of our work and for recommending publication in Nature Communications.

1) Several important quantities, such as the "two-sided spectral densities" S_{imp} , S_{phi} and the feedback S_{BA} , are not defined in the text.

In the revised manuscript, we added an explanation of these spectra in the end of the first paragraph of Section II "Measurements of spin oscillators".

We note that "two-sided" refers to the normalization convention of power spectral densities, as is now explained in SI Section C, with an appropriate reference from the main text. See also the response to question 3 of Reviewer 1.

2) Equation (4) is important as it describes the deviation of the interaction from the simple form $X_L * X_S$ due to the presence of various levels in the ground atomic state. I think the authors should explain where and under what assumptions this formula was derived, pointing to the important equations and sections in the Supplementary Information or other references.

To summarize, the key approximations that go into equation (4) are

- a) low saturation of the atomic transitions
- b) a low enough density of atoms so that photons incoherently scattered within the ensemble leave the ensemble without interacting with the atoms second time
- c) macroscopicity of the number of atoms – that the relative changes in the populations of the atomic levels produced by the vacuum fluctuations of the probe light are negligible.
- d) the coherent amplitude of the probe light is large compared to its fluctuations, and does not change as the light propagates through the cell

Essentially, equation (4) in its present form was derived in the SI, which also contained the relevant references we are aware of.

In the revised manuscript, before proving Equation (4), we state that it is derived in the SI and re-cite the references from the SI. We also put in the SI, the beginning of Section B, the above list of assumptions underlying our theoretical treatment.

3) Several times in the article the notion of thermal occupation of oscillator levels appears. I do not quite understand what this thermal distribution corresponds to, how it is obtained, and what the corresponding temperature depends on. If the answer is to be found in the supplementary material, the authors should indicate a precise reference to the section and equations.

The "temperatures", T_m , of the oscillator *baths* (not the oscillators themselves, because they are generally not in equilibria with their baths) parametrize the occupancies N_m of the magnetic sublevels of atoms in the steady state, which are determined by the rates of the spontaneous scattering of the probe and repump phonons. The temperatures are defined for each transition $m \rightarrow m + 1$ as

$$\exp\left(\frac{k_B T_m}{\hbar \Omega_m}\right) = \frac{N_{m+1}}{N_m},$$

where Ω_m is the frequency of the transition between the magnetic moment projections m and $m + 1$ (Eq. SI B 19).

There are separate temperatures for different m , which in principle do not need to be equal, but experimentally we find that they are all equal within the uncertainty with which we can characterize them, hence we speak about only one temperature.

What justifies the introduction of such a temperature is that it describes the numbers of excitations in the collective modes $\{X_m, P_m\}$ when the quantum backaction of measurements is negligible (i.e., in the low-coupling limit, in a situation analogous to thermal equilibrium), and also it enters the rate at which the coherences between different magnetic sublevels decay.

Note that the distribution of the atoms over magnetic sublevels can also be exponential for other relaxation mechanisms than spontaneous scattering, such as collisions (see Appelt et al. "Theory of spin-exchange optical pumping of ^3He and ^{129}Xe ." Phys. Rev. A, 58, 1412,1998).

In the revised supplementary, we separated the discussion of the temperature of the intrinsic damping baths in a subsection of section B, and, in the revised manuscript, added a reference to this subsection when mentioning the thermal occupation of the optical baths experienced by the oscillators.

Reviewer #3:

In their manuscript, Baerentsen et. al. look at the dynamics of an atomic spin probed by light, and investigate how the Faraday interaction results in squeezing of the effective spin. This is done for different regimes, characterised by the ratio between the measurement rate and the spin oscillator frequency.

The study is potentially interesting, but the text is extremely hard to follow even for someone with a bit of experience in QND-based squeezing, and light-matter Faraday interaction. The language used is very technical and it requires knowledge of previous experiments from the group. Moreover, it is often lacking a clear and intuitive explanation of the physics behind the effects investigated.

We thank the referee for the constructive criticism and feedback on our work. Our intention when writing the manuscript was to make it also accessible to people from other communities than atomic physics, such as levitated nanoparticles, gravitational wave detection, and optomechanics, and, as tradeoff, our terminology may at times appear unusual for someone with expertise in atomic physics. In the revised manuscript and supplement, we tried making the narrative more self-contained, and to explain more comprehensively the concepts we invoke. The particular edits we made are covered in the answers to the specific points of this and other reviews.

One point we need to clarify from the beginning is that the focus of our work is the squeezing of light. While our data give indirect evidence that there is also a squeezing of the spin, verifying this fact directly would require quite different measurements, which we did not perform.

I list below a number of points that I found particularly confusing, and that should be better addressed or explained in the text.

-- in the introduction, why the authors mention POVMs rather than QND measurements? the link between these two concepts should be clarified to the reader.

The measurements that we perform on the atomic spin are not, generally speaking, of QND type because the spin precesses in the magnetic field. It is only when the precession is much slower than the rate of measurements they would become QND. In our experiment, we only access an intermediate regime where the rates of the precession and measurement are of the same order. Such measurements are described by POVMs, as they do not project the spin state on a state from an orthogonal basis.

-- when talking about position measurement of the oscillator, it should be mentioned in which frame this is performed. I guess the lab frame.

We thank the Reviewer for pointing out this ambiguity. Correct, the position is in the lab frame, not the rotating frame of the precession. In the revised manuscript, we mentioned this in the first paragraph.

-- "When the rate is slower than the oscillation, measurements with the meter in the vacuum input state project the oscillator on coherent states." is this simply because when measuring the position of the oscillator in the lab frame for long time results in an averaging over a "rotating quadrature"? So there is a continuous squeezing and anti-squeezing as the system evolves?

Indeed, the fact that a slow continuous measurement projects the spin oscillator on a coherent state can be understood as being due to the averaging over the quadrature rotation. Because of the averaging, the spin state should be rotationally symmetric in the phase space.

There is a continuous squeezing of the meter light as the measurement happens, while the spin remains in a coherent state, whose coherent displacement diffuses over time.

-- how does this measurement rate faster than the oscillation compare to the stroboscopic measurement approach that was also developed by the same group? If the interaction is weak, which results in a slow measurement rate, one can always decompose the measurement into shorter and stroboscopic pulses. Is this correct? What would be the difference (e.g. in terms of bandwidth)?

This is a very good question, which we don't have a simple answer to. It is not correct that a continuous measurement in any limit reproduces a stroboscopic measurement. While stroboscopic measurements are phase-sensitive (they are "blind" to the Fourier components of the external forces that are out of phase with the probing pulses), continuous measurements are phase-insensitive. The bandwidth of a stroboscopic measurement is determined by the peak probe power, the same as for a continuous measurement. In terms of the generation of spin squeezing, it is difficult to tell which approach would work better in the same setting, and this can be a subject of further investigation.

-- what does it mean "the rate of position measurement affects the output state of the meter field"? Is it really the rate (i.e. the "speed") that affects the field? Or the authors are just referring to the measurement back-action?

We only refer to the fact that the spectrum of the interference between the measurement back-action and imprecision looks qualitatively different depending on the measurement-rate-to-precession-frequency ratio.

In the revised manuscript, we rephrased the sentence as "The output state of the meter field changes with the measurement regime as much as the oscillator state itself". We hope that this wording is less confusing.

-- when saying "In the slow measurement regime, the correlations and the associated squeezing exist in a narrow frequency band near the resonance, and have a strong frequency dependence due to the time-averaged response of the oscillator to the measurement backaction", what does it mean the strong frequency dependence? The sentence already says squeezing exist in a narrow frequency band, so is this a redundant statement or the authors mean something else that is not explained?

We thank the Reviewer for raising this point. There was indeed a redundancy in this sentence. In the revised text, we removed the clause "and have a strong frequency dependence".

-- The author should give a clear physical and intuitive picture of why for measurement rates faster than the oscillation the squeezing of the field is broadband.

The squeezing bandwidth is the range of optical sideband frequencies for which the nonlinearity of the atomic medium is high enough to modify the variance of the optical states at the scale of vacuum fluctuations. This nonlinearity is resonantly enhanced around the Larmor frequency. In the case of negligible spin decoherence, no matter how weak the probe power is (i.e. how slow is the measurement rate), there is a correspondingly small detuning from the Larmor resonance where the interaction is resonantly enhanced enough to produce optical squeezing.

Generally, the range of detunings around the resonance that produce squeezing is approximately proportional to the measurement rate, regardless of whether this rate is fast or slow. Therefore, as the measurement rate reaches the oscillation frequency, so does the squeezing bandwidth, and the squeezing becomes broadband.

In the revised manuscript, we added a short version of this explanation in the middle of Section II.

-- Is it true that in the slow measurement regime the oscillator is projected into a coherent state, but the light field get squeezed? This is not explained clearly.

Yes, this is true. We hope that the extended explanation of the squeezing bandwidth mentioned in the answer to the previous question makes it more clear that the light always gets squeezed, with just the properties of squeezing being different in the different measurement regimes.

-- "quantum fluctuations of ponderomotive torque" is not explained.

Radiation pressure forces are commonly referred to as "ponderomotive forces", and the squeezing of light resulting from the measurement back-action imparted by such forces is called "ponderomotive squeezing". For example, see our Ref [19-21], and the historical paper by Braginsky and Manukin "Ponderomotive Effects of Electromagnetic Radiation", JETP 25, 653 (1967). Since in our experiment the polarization quadratures of light change the angular momentum of the atomic ensemble by applying torque to it, we called this effect "ponderomotive torque".

To address the concern raised by the reviewer, in the revised manuscript, we refrain from using this newly introduced term and write "quantum fluctuations of **optical** torque" instead, which should be more understandable.

-- "The Heisenberg uncertainty principle constrains the two-sided spectral densities of the imprecision in ...", it is not clear. Someone not familiar with the connection between spectral density and oscillator

dynamics is required to read extra literature and SI, but this measurements are crucial to perform quadrature measurements and should be explained in the text for the article to be self-contained.

We thank the Reviewer for pointing out the lack of completeness. In fact, this constraint on the spectral densities has little to do with the oscillator dynamics. It is better described as an error-disturbance uncertainty relation for the quantum measurement (For a high-level definition, see M. Ozawa, “Universally valid reformulation of the Heisenberg uncertainty principle on noise and disturbance in measurement”, *Phys. Rev. A* 67, 042105, 2003).

In the revised manuscript, we added an explanation of the imprecision and backaction spectra in the beginning of Section II which, hopefully, makes the origin of the inequality more straightforward and shows that no oscillator dynamics are involved.

We note that the general explicit definitions of S_{imp} and S_{BA} , given in SI Section C, are quite complex, because they include all conceivable classical noise contributions, which are, moreover, specific to our experimental configuration. For this reason, we believe that including them in the main text would reduce the clarity rather than improve it. We did, however, add an extra reference to SI Section C in the main text.

-- How is Eq.2 derived? Details should we put in the SI.

To address this comment, we added a step-by-step derivation of Eq. 2 as a separate subsection to SI Section F. We also note that Eq.2 is simply Eq. SI F10 (F1 in the former version) with ζ set to 0.

-- Is the term " $P_m \cdot PL$ " in Eq.4 coming from tensor polarisability? The term "dynamical backaction." is not explained, and is not clear what it means. Missing citations to previous works of the group where this Hamiltonian is derived in detail.

Yes, the term " $P_m \cdot PL$ " comes from tensor polarizability. The Hamiltonian in exactly the form given in Eq.4 is derived in SI section B, as mentioned explicitly in the revised manuscript immediately before Eq.4. To highlight the prior works more clearly, in the revised main text we also re-cited the relevant to the derivation of Eq.4 references from the SI.

Dynamical backaction is the modification of the oscillator damping rate and resonance frequency due to its coupling to a measuring device (the forward scattering modes of light in the present case). This term emerged in the literature on continuous position measurements (dating back to at least the book of VB Braginsky and AB Manukin “Measurement of Weak Forces in Physics Experiments”, University of Chicago, 1977), and is widely used in the modern literature on optomechanics, magnomechanics, and levitated nanoparticles (as in, e.g. Schliesser et al. “Radiation Pressure Cooling of a Micromechanical Oscillator Using Dynamical Backaction”. *Phys. Rev. Lett.* 97, 243905, 2006.).

In our early work involving interaction of forward scattered light with atomic ensembles we described such an effect by a sum of a “beam splitter” and an “entangling” interactions in the Hamiltonian with unequal weights, that is by a single term version of eq. (4) of the present paper. We have added references to our previous papers where this type of Hamiltonian including the tensor polarizability term has been used [Wasilewski et al, **Optics Express** 17, 14444-14457 (2009), Krauter et al. **Phys. Rev. Lett.** 107, 080503 (2011), Muschik et al, *J. Phys. B: At. Mol. Opt. Phys.* 45 124021 (2012)].

In the revised supplementary information, we added a dedicated subsection to Sec. B that explains the term dynamical backaction, its origin, and its relation to another effect that modifies the damping rate of the spin known as “power broadening”. In the revised main text, we refer to this part of the SI when mentioning dynamical backaction.

-- what does it mean "The multimode structure can affect the response of the spin to the measurement backaction at frequencies close to Ω_S "? Affect in which way? Why?

“Affects” in this context means modifies the response of the spin ensemble to a generalized force driving it (classical magnetic excitation or quantum backaction).

This is essentially the consequence of the frequency splittings and the coupling coefficients being small compared to the Larmor frequency Ω_S . At large frequency offsets $\Delta\Omega$ from Ω_S , the effect of the splittings and couplings is a small perturbation that goes to zero as $\Delta\Omega$ increases.

In the revised manuscript, we largely rewrote Sec II, which, hopefully, makes it convey the information more clearly.

-- when discussing the population of the F sub-levels, can the authors give quantitative numbers? What fraction of atoms occupy each of the m_F levels?

We thank the reviewer for raising this point. These numbers in fact can be inferred almost exactly from the value of the polarization, because the distribution of atoms over the magnetic sublevels is close to exponential.

The relative populations of the m_F levels are (0.52, 0.23, 0.12, 0.06, ...). Note that the fractional errors for the individual numbers can be 10-20 %, while the average polarization is known with a better precision, as $p = 0.78 \pm 0.04$.

In the revised supplementary material, Section A describing the experiment, we now dedicate a paragraph explaining that the distribution of atoms over the magnetic sublevels is approximately exponential and give the parameters of this distribution.

-- "We work in the negative-mass configuration", why? what would be different with a positive mass oscillator (ie opposite field orientation)?

This is mainly for historical reasons that we work with the negative mass configuration – the compensation currents ensuring the magnetic field homogeneity and the polarization of the repumping light were optimized for one particular orientation. There is no big difference between the positive and negative mass configurations, except for the overall sign of the correlations, as mentioned in SI section E and illustrated in SI Fig 2d.

-- For all figures, it is missing the experimental sequence used to take the measurements. Example: Fig.1d, is the result of only one light pulse continuously measured? How long? Or there is one state-preparation pulse, followed by a read-out pulse? Is there any operation done between consecutive shots to "reset" the system? Or just waiting?

Similar description of the sequences are needed for each figure.

All the data we present in the manuscript were taken in steady state – all the lasers, magnetic fields, and detectors were continuously on. There were no sequences or pulses. We did use pulsed Magneto-Optical Resonance Signal (MORS) method to characterize the spin polarization we report, but this is a very well-known and technical procedure (which we nevertheless added references to, both in the main text and in the SI).

-- "From a global fit of the spectra at all quadratures", with what function? Eq.2?

With equations SI D 1-3, which generalize Eq. 2. In the revised manuscript, we write this explicitly:

“From a global fit of the spectra at all quadratures using an expression that generalizes Eq. 2 to the case of finite ζ and extra detection noise (see SI Sec. D), we infer ...”

-- The emergence of bright modes is discussed on the right of Eq.5 only very qualitatively. This part should be better explained and made more rigorous. The merging of the eight peaks into two and three broader peaks is neither explained nor modelled, and an intuitive picture of the physical process is completely missing.

We thank the reviewer for raising this interesting point. In this particular case, we have to be satisfied with a qualitative explanation ourselves, because our attempts to model the hybridization from the first principles were only a limited success.

While the model we use to fit our data in Fig 2 and 3 (which was described in SI Section D) is based on an empirical assumption that the complex dynamics of the spin can be explained as arising from only two or three interacting oscillator modes with some effective parameters, we could instead calculate the output spectrum based on the full 16-mode first-principle model presented in SI Section B.

To give an idea how good this model agrees with the experimental data, we provide the comparison for one case in the figure below. On the left, we reproduce Figure 2a from our manuscript, where the colored dots show experimental data, and the solid back lines show the fit using the model involving two interacting harmonic oscillators and an incoherent background due to the broadband noise (as described in SI Section D).

On the right, we calculate the spectra based on the more complete model, where the only interaction parameter that is not independently calibrated is the overall measurement rate of the spin. One can see a fair agreement between the theoretical model and the experimental data. The theoretical model correctly predicts the emergence of two hybridized modes, a and b , and a few linewidths away from the spin resonance the theoretical curves are almost indistinguishable from the experimental.

Resp. Fig. 1. The emergence of two peaks in Fig 2a.

Based on the simulation results, one can enquire what exactly a combination of the atomic oscillators contributes to each peak, and the answer we are getting that peak *b* is mostly due to the oscillator $m_F 3 \leftrightarrow 4$, and the peak *a* is a complex combination of all others.

It is also clear, however, that our first-principle model does not reproduce all the details of the noise structure around the spin resonance (the Larmor frequency) – the peak linewidths, amplitudes, and splitting are slightly different in experiment from what the theory predicts. We hope to investigate more thoroughly in the future what cause those differences, and possibly make it a subject of a separate work, but with the results we have now, we consider it to be beyond the scope of the manuscript to discuss the hybridization further.

To summarize, to the best of our understanding, the qualitative explanations of hybridization that we provide have solid theoretical foundation, but we would not like to present more quantitative results until our understanding is more complete.

To address the Reviewer’s concerns, we reproduced part of the explanation of the modelling procedure given in the answer above in the beginning of Sec D of the revised supplement. We hope that this makes it more clear what assumptions went into the model and what was the motivation for introducing them.

-- As a curiosity, how is it achieved shot-noise-limited readout with a power of $\sim 12\text{mW}$? What are the details of the detector, and of the readout setup? What is the minimum field quadrature noise that could be measured (ie maximum squeezing)?

The shot-noise limited readout at 12 mW power is enabled by two features of our experiment.

First, in the balanced polarization homodyning setup that we are using neither the amplitude nor the phase noise of the laser contributes to in the detected signal – any amplitude noise is suppressed by the power balancing of the beams on the two photodetectors, while the phase noise is suppressed because the coherent component of the laser beam that serves as the LO co-propagates along the same optical path with the signal field.

Second, at the polarization of the probe that we are using, the amplitude or phase laser noises also do not drive the atomic ensemble (see the discussion of classical noises in “Entanglement and Quantum Interactions with Macroscopic Gas Samples.” Julsgaard, PhD thesis, 2003).

The degree of the suppression of the amplitude laser noise depends on the purity and alignment of the input polarization, and the orientation of the homodyne waveplates. It does take some amount of empirical tuning of these parameters to cancel the classical noises in the detection well enough.

-- Details on the derivation of Eq.6 are missing.

In the revised manuscript, we write that Eq.6 is obtained from Eq.2 by setting $\phi = 0$, $\Omega = 0$, and using the approximation $\Gamma \gg \gamma_{th}$.

-- I did not find clear how to understand the peaks in Fig.2b, and the energy spectrum on the inset. Are these the 2F modes coming from performing the HP transformation of each angular momentum?

Each of the 2F peak in the lowest-power trace of Fig. 2b corresponds to one transition between the neighboring mF levels of F=4, not the mF levels themselves. The energy splittings between the neighboring mF levels are slightly different because of the quadratic Zeeman and tensor Stark shifts (as was meant to be illustrated by the inset in Fig.2b), which is why the frequency of the peaks are slightly different from each other. The formal definitions of the corresponding modes are given in the SI. Indeed, the excitations of these modes can be thought about in the framework of HP approximation applied to the individual two-level systems formed by the levels involved in the separate transitions.

-- what is the physics behind these transitions between m_F levels? How does it happen? Why classically driven motion does not simply result in a coherent state of the modes defined through the HP transformation?

The transitions between the neighboring mF levels can be due to Raman scattering of the probe photons (as usual for Faraday interaction), or classical excitation by the magnetic field. The HP transformation, the way it is usually made, does not assume that there is a macroscopic number of atoms in any of the excited mF levels, as we have it in our experiment with finite spin polarization.

When a classical magnetic field is applied to drive the spin ensemble, it does create coherent states in each of the oscillator modes. The traditional HP situation can be seen as a limit in which the interaction strengths of the probe with all modes but one (in which the atoms jump between the outer-most mF=4 and its adjacent mF=3) go to zero.

-- The coupling between the modes, as a function of the magnetic field, should be better explained and investigated. How the coupling mentioned in the text above Eq.6 depends on the detuning? It is not explained the physical mechanism behind this.

The coupling coefficients between the modes should not depend on the magnetic field, while the splitting of the mode frequencies grows linearly with the magnetic field, so the hybridization effects should impact the signal spectrum less at high magnetic fields. Unfortunately, we have limited experimental capabilities to investigate this dependence, because the low-noise current source we use to create the magnetic field is not powerful enough to shift the Larmor frequency above 2 MHz.

The dependence of the coupling between the modes on the probe detuning Δ is complicated in details, but the coupling magnitude roughly scales as $1/\Delta$ if the light-spin interaction cooperativity is fixed. (And if it is the power of the probe laser that is fixed, the coupling scales approximately as $1/\Delta^3$, because the measurement rate scales as $1/\Delta^2$, and the coupling coefficients are products of ζ s and the

measurement rates). The full dependence of the coupling on the detuning is encoded in Eq. SI B 23-24 through the detuning dependence of ζ s.

In the revised manuscript, we comment on the optical detuning dependence of the coupling between the spin modes in the same paragraph where we describe the data shown in Fig. 2b.

Reviewer #1 (Remarks to the Author):

The authors have well addressed all my comments and questions in great details. I have also read the authors' response to the other two referees' and in my view the answers are clear and sound. Since the result in this work is new, of high quality, and also important to many fields in physics, I recommend publication of this thoroughly revised manuscript.

Reviewer #2 (Remarks to the Author):

The authors answered my questions satisfactorily and improved the main text and additional information. I therefore recommend publication of the article.

Reviewer #3 (Remarks to the Author):

I thank the authors for addressing my questions.
I suggest the work for publication.